

# Daylight savings time transition and the incidence of femur fractures in the older population: a nationwide registry-based study

Ville Ponkilainen[1], Topias Koukkula[2,3], Mikko Uimonen[1], Ville M. Mattila[2,3], Ilari Kuitunen[4,5] and Aleksi Reito[2,3]

[1] Department of Surgery, Central Finland Hospital, Jyväskylä, Tampere, Finland
[2] Center for Musculoskeletal Diseases, Tampere University Hospital, Tampere, Finland
[3] Faculty of Medicine and Health Technology, Tampere Univeristy, Tampere, Finland
[4] Institute of Clinical Medicine, University of Eastern Finland, Kuopio, Finland
[5] Emergency Department, Mikkeli Central Hospital, Mikkeli, Finland

Corresponding author
Ilari Kuitunen, ilari.kuitunen@uef.fi

## ABSTRACT

**Background.** Daylight Savings Time (DST) transition is known to cause sleep disruption, and thus may increase the incidence of injuries and accidents during the week following the transition. The aim of this study was to assess the incidence of femur fractures after DST transition.

**Methods.** We conducted retrospective population-based register study. All Finnish patients 70 years or older who were admitted to hospital due to femur fracture between 1997 and 2020 were gathered from the Finnish National Hospital Discharge Register. Negative binomial regression with 95% confidence intervals (CI) was used to evaluate the incidence of femur fractures after DST transition.

**Results.** The data included a total of 112,658 femur fractures during the study period between 1997 and 2020, with an annual mean (SD) of 4,694 (206) fractures. The incidence of femur fractures decreased at the beginning of the study period from 968 to 688 per 100,000 person-years between 1997 and 2007. The weekly mean of femur fractures remained lower during the summer (from 130 to 150 per 100,000 person-weeks) than in winter (from 160 to 180 per 100,000 person-weeks). Incidence rate ratio for the Monday following DST transition was 1.10 (CI [0.98–1.24]) in spring and 1.10 (CI [0.97–1.24]) in fall, and for the whole week 1.07 (CI [1.01–1.14]) in spring and 0.97 (CI [0.83–1.13]) in fall.

**Conclusion.** We found weak evidence that the incidence of femur fractures increases after DST transition in the spring.

# INTRODUCTION

Daylight Savings Time (DST) transition represents the act of turning the clocks and it is applied in at least 70 countries worldwide. DST has been shown to cause sleep disruption, fragmentation of the circadian rhythm, and fatigue after the transition (*Lahti et al., 2006b*;

*Lahti et al., 2006a*; *Lahti et al., 2008*). Moreover, these minor jet lag symptoms are known to decrease attention and alertness for multiple days after the transition (*Lahti et al., 2006b*; *Lahti et al., 2006a*; *Lahti et al., 2008*).

It has been hypothesized that sleep disturbance leads to changes in sympathetic activity which, in turn, increases heart rate and blood pressure, and thus more proinflammatory cytokines are released (*Rishi et al., 2020*). The combination of these actions may lead to increased incidence of cardiovascular events such as acute myocardial infarctions (AMI) and cerebral strokes. It has been further hypothesized that the sleep disruption caused by DST leads to higher incidence of motor vehicle collisions than normal, yet the literature is contradictory (*Carey & Sarma, 2017*; *Prats-Uribe, Tobías & Prieto-Alhambra, 2018*). In addition, DST does not seem to increase trauma admissions to the ED (*Lee, Stahlman & Sharrah, 2019*).

Femur fractures are the most common fractures in older adults and are often caused by low-energy falls and tumbles (*Jantzen et al., 2018*; *Reito et al., 2019*; *Meyer et al., 2021*). Moreover, femur fractures cause high morbidity and mortality, and the societal costs of these injuries are high (*Gabriel et al., 2002*; *Hernlund et al., 2013*; *Jang et al., 2021*). As these fractures often occur to the most fragile individuals, it can be hypothesized that changes in the circadian rhythm may lead to increased incidence following DST. Thus, the aim of this study was to assess the incidence of femur fractures after the transition to DST and back in the population 70 years and older in Finland.

## MATERIALS & METHODS

The data were based on the Finnish National Hospital Discharge Register (NHDR). All femur fractures incurred in Finland between 1997 and 2020 were included in the study using the ICD-10 (Finnish version) codes S72.0–S72.4. All patients had also undergone any femoral fracture fixation or replacement. The study population comprised all inhabitants 70 years and over, approximately 800,000 persons in Finland (population in 2020). Only patients older than 70 years of age were included, as the percentage of low-energy femur fractures is more common among this group. The primary outcome, femur fracture, was collected by including chronologically the first treatment period of each patient for the corresponding code. All patients which had femur fracture as a primary or secondary diagnoses were included. The date of the injury was the date that the patient was admitted to hospital. The coverage and accuracy of the dates, diagnoses, and procedure codes of NHDR are high (*Mattila et al., 2008*; *Sund, 2012*). However, the limitation of the NHDR is that it does not contain information regarding the laterality. Therefore, if the patient had two femur fractures, only the first was included in the study.

In Finland, DST transition takes place on the last Sunday of March and the transition back to the normal time on the last Sunday of October. However, due to distances and delay between the fall and operation, we evaluated the incidence of hip fracture surgery during the following week. In this study, the DST period was investigated using models that included the first Monday and the first week after DST transition. We investigated DST transition both in spring when the transition is forwards, resulting in one hour less sleep, and in the fall when the transition is backwards, resulting in one hour more sleep.

## Statistical analysis

The analyses were conducted by using negative binomial regression to adjust for seasonal variation. Periodical explanatory variables were factorized weeks (1–52), weekdays, years, and DST as a binary variable to account for the normal seasonal variation. The daily incidence of hip fractures was the dependent variable. Incidence can also be considered as a count data. This means that a single day (*e.g.*, May 4th, May 5th, May 6th) is the study unit or case, and each day has a specific incidence as described. The analyses were performed using two models: (1) model for the following Monday adjusted by year and week as factors, (2) model for the next week adjusted by year, week, and weekday. Both models involve daily incidence as an outcome variable. In the first model, the DST variable is the main exposure coded 1 for the first Monday after spring or fall transition and 0 for all other days. This model includes factorized year and week as covariates meaning that each day has an associated year and week, and the year 1997 and week 1 were used as reference categories in factorized variables. The value of the regression coefficient for DST was interpreted as the main effect of interest or the incidence rate ratios (IRR) for the Monday after transition in spring and fall. The second model was similar, except that the whole entire week after the transition was coded 1 for the DST variable and covariates included were a year, week and weekday (Monday as reference). The Pearson residual scale was used to evaluate the goodness-of-fit of the model. The results of the models were interpreted as IRR with 95% confidence intervals (CI), as comparison between DST and non-DST Mondays and weeks. Degrees of freedom (df) and McFadden's pseudo-$R^2$ were reported for each model. For sensitivity analyses we also report IRRs using two different Poisson models. First model include incidences as an outcome variable similar to negative binomial regression and second model includes daily absolute count as an outcome variable and yearly population as an off-set variable. Further steps taken in the Poisson regression were similar to negative binomial regression. Similar performance measures are reported for the Poisson models.

The annual age-adjusted incidence (per 100,000 person-years) was calculated based on the corresponding age population in Finland obtained from the national population register (Official Statistics of Finland). Reference population was the mean of all citizen older than 70 years during the study years. All analyses were performed using R version 4.0.5 (R Foundation for Statistical Computing, Vienna, Austria). Ethical committee evaluation was not needed due to the retrospective and register-based design. This study has been reported according to the strengthening the reporting of observational studies in epidemiology (STROBE) and the checklist is provided as File S1.

## RESULTS

A total of 112,658 femur fractures were included. Women had 74.3% of the fractures (Table 1). The highest age groups sustaining fractures were inhabitants aged 80–84 (29431 fractures) and 85–89 (29235 fractures) years (Table 1). The incidence of femur fractures decreased at the beginning of the study period from 968 to 688 per 100,000 person-years between 1997 and 2007 (Fig. 1). The weekly mean of femur fractures remained lower during the summer (from 130 to 150 per 100,000 person-weeks) than in winter (from 160 to 180 per 100,000 person-weeks) (Fig. 2).

| Table 1 | Background characteristics of included fracture patients. | |
|---|---|---|
| | *n* | % |
| Age group | | |
| 70–74 | 13,538 | 12.2 |
| 75–79 | 21,415 | 19.3 |
| 80–84 | 28,848 | 26.1 |
| 85–89 | 28,496 | 25.7 |
| 90–94 | 14,828 | 13.4 |
| 95–99 | 3,305 | 3.0 |
| 100 or more | 303 | 0.3 |
| Gender | | |
| Male | 28,102 | 25.4 |
| Female | 82,631 | 74.6 |

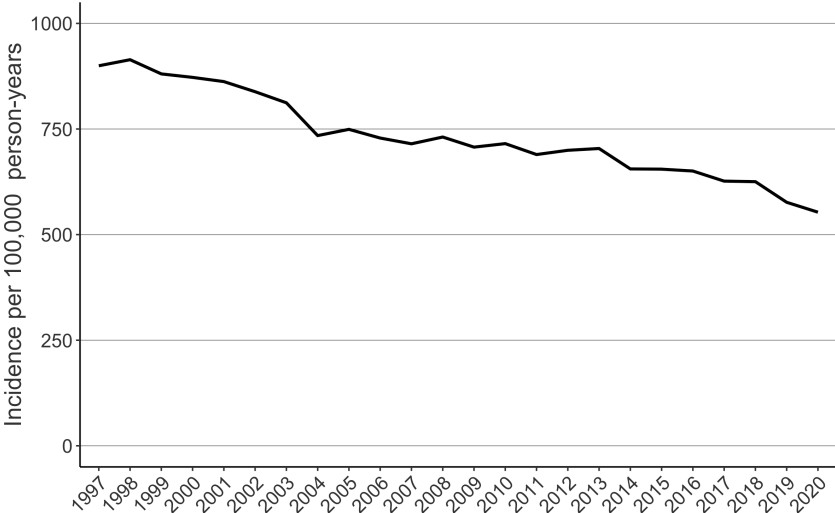

**Figure 1** The age-adjusted incidence of femur fractures in patients 70 years or older in Finland between 1997 and 2020.

Based on the multivariable negative binomial regression model, we observed an 10% increase in femur fracture incidence on the Monday following DST transition in spring (IRR: 1.10 CI [0.98–1.24], $df = 25$, pseudo-$R^2 = 0.16$). In the fall, we observed an 10% increase (IRR: 1.10 CI [0.97–1.24], $df = 25$, pseudo-$R^2 = 0.16$). When looking at the whole week following DST transition in the spring, we observed a 7% increase (IRR: 1.07 95% CI [1.01–1.14], $df = 82$, pseudo-$R^2 = 0.26$). In the fall, we observed a 3% decrease in the incidence (IRR: 0.97 95% CI [0.83–1.13], $df = 82$, pseudo-$R^2 = 0.26$). Results were similar when Poisson models was used (Table 2).

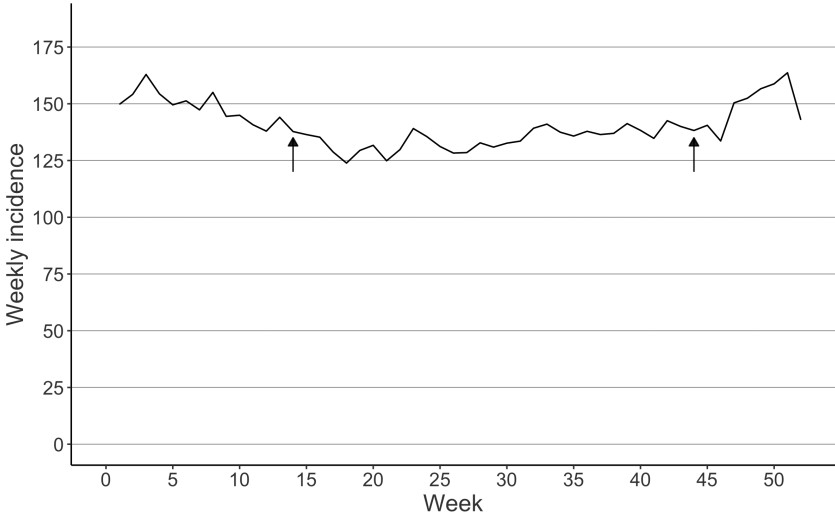

**Figure 2** **Seasonal variation interpreted as the mean incidence of femur fractures per week in Finland between 1997 and 2020.** Arrows indicate the DST transition in spring and fall.

**Table 2** **Results of the two different Poisson models.**

|  |  | Poisson model 1 | McFadden pseudo-$R^2$ | Poisson model 2 | McFadden pseudo-$R^2$ |
|---|---|---|---|---|---|
| Monday following DST | Spring | 1.10 (1.01–1.20) | 0.16 | 1.10 (0.99–1.23) | 0.17 |
|  | Fall | 1.11 (1.01–1.20) | 0.16 | 1.09 (0.98–1.21) | 0.17 |
| Whole week following DST | Spring | 1.07 (1.02–1.12) | 0.27 | 1.07 (1.01–1.13) | 0.27 |
|  | Fall | 0.97 (0.86–1.09) | 0.27 | 0.97 (0.83–1.13) | 0.27 |

# DISCUSSION

In our study, we observed a higher incidence of femur fracture on the Monday following DST transition, adjusting for other temporal factors, but we were unable to show clear evidence against the hypothesis that incidence would remain the same after DST transition. Temporal variables explained surprisingly high proportion of all observed variability (16–27%). When the week following DST transition was analyzed, we observed a higher incidence of femur fractures and found scarce evidence against the hypothesis that incidence would remain same after DST transition. As a whole, our study showed weak evidence that femur fracture incidence in older patients increases after DST transition, especially in the spring.

Consequences of DST has been investigated widely. In physiological level DST has been associated altered cortisol levels, increased levels of plasma pro- and anti-inflammatory markers and lower vagal tone (*Grimaldi et al., 2016*; *Rishi et al., 2020*; *Wright Jr et al., 2015*). In patient level these may manifest as sleep deprivation and misaligned circadian rhythm (*Lahti et al., 2008*; *Rishi et al., 2020*). In population level one of the most investigated topic has been the incidence of acute cardiovascular events following DST. Numerous

studies suggest that incidence of AMI is significantly higher during the week following the DST (*Sandhu, Seth & Gurm, 2014*; *Sipilä, Rautava & Kytö, 2016*). Effect of DST to cardiovascular deaths is unclear (*Manfredini et al., 2019*). Number of hospital admissions due to atrial fibrillation has been also associated to DST in the spring time especially in female patients (*Chudow et al., 2020*). DST has shown also increased incidence of ischemic strokes (*Malow, Veatch & Bagai, 2020*; *Sipilä et al., 2016*). In general, it is evident that DST is associated to several adverse cardiovascular health effects in the population level.

To the best of the authors' knowledge, this was the first study to investigate the influence of DST transition on femur fracture incidence. It has been previously hypothesized that there are more motor vehicle collisions on the Monday following DST transition than normal, although the results have been questioned in a recent systematic review (*Carey & Sarma, 2017*; *Prats-Uribe, Tobías & Prieto-Alhambra, 2018*). In addition, the transition does not seem to increase the number of trauma admissions (*Lee, Stahlman & Sharrah, 2019*). Our hypothesis was that the DST may still affect most fragile elderly citizen and thus lead to increased risk of falling. The most common fall related fractures were considered to include to most comprehensive data possible. As a result, it seems that the one-hour reduction or increase in sleeping time in the spring and fall is weakly seen in the number of falls and subsequent rates of femur fractures in the older population.

Patients sustaining a femur are generally frail. Femur fractures are also seen in younger patients due to high energy trauma such as motor vehicle collisions and falls from height. Very large majority of femur fracture are, however, seen in older patients who have fallen in the same level. Several known risk factors have been associated to falls and subsequent femur fracture in the older patients. These include muscle weakness, impaired walking ability, impaired eye-sight and poor nutritional status. Suggested physiological effects of DST include sleep distruption, circadian misalignment and lowered vagal response. If these effects are truly endemic it is then very likely that they also increase the risk of fall and subsequent femur fracture in patients who have known risk factor for a fall.

The strength of this study is the nationwide register data that include all femur fractures between 1997 and 2020. Our study has some limitations also. Firstly, it is possible that the admission date in the register is not the same as the injury date, in case if there was a delay between the trauma and admission to hospital, or if the patients were admitted to hospital shortly after the day changed. Femur fracture, however, leads to immediate inability to ambulate and walk and usually these patients are transported quickly to hospital. Secondly, some bias may always be due to coding errors as the main data source is a hospital discharge database. It is worth mentioning, however, that FHDR has shown high validity in terms of completeness and accuracy (*Huttunen et al., 2014*; *Sund, 2012*).

## CONCLUSIONS

We found weak evidence that the incidence of femur fractures increases after DST transition in the spring.

### Funding

The authors received no funding for this work.

### Competing Interests

The authors declare there are no competing interests.

### Author Contributions

- Ville Ponkilainen performed the experiments, analyzed the data, prepared figures and/or tables, and approved the final draft.
- Topias Koukkula performed the experiments, analyzed the data, prepared figures and/or tables, contributed data, and approved the final draft.
- Mikko Uimonen conceived and designed the experiments, authored or reviewed drafts of the article, and approved the final draft.
- Ville M. Mattila performed the experiments, authored or reviewed drafts of the article, contributed data, and approved the final draft.
- Ilari Kuitunen conceived and designed the experiments, authored or reviewed drafts of the article, and approved the final draft.
- Aleksi Reito conceived and designed the experiments, performed the experiments, analyzed the data, authored or reviewed drafts of the article, and approved the final draft.

### Data Availability

The code for the analyses is available in the File S1.

The de-identified study data were provided for peer review but access to the data requires an application to the Finnish National Hospital Discharge Register. Please contact the Finnish Social and Health Data Authority at https://findata.fi/en/permit/.

Data can be found https://aineistokatalogi.fi/catalog/studies/7567e45d-72b7-428b-be9e-510440336edf.

Key variables in full dataset, which we used in our analysis and limitation was made:

ICD10O_1-26 (26 variables) = Diagnose code given during the treatment period (ICD-10 diagnose code classification).

-We defined the diagnose code limitation S72.0–S72.4 based on those variables.

TUPVA = Patient check-in date and time (=date of injury).

-We defined year, week, weekday, DST-time and all seasonal variables based on this variable. We also defined year exclusion based on this variable.

IKA = Age of the patient.

-We defined our age group exclusion based on this variable.

After exclusions, the actual count data used in analysis (ie. negative binomial regression) was formed based on TUPVA variable.

## Supplemental Information

Supplemental information for this article can be found online at http://dx.doi.org/10.7717/peerj.13672#supplemental-information.

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
