# Peer review of "Daylight savings time transition and the incidence of femur fractures in the older population: a nationwide registry-based study"

_PeerJ, doi:10.7717/peerj.13672_

## Round 0.1 · original submission · Major Revisions

Thank you very much for your submission. While your paper represents an important contribution to the area of the influence DST has, your paper has a number of areas in which our reviewers feel it must be improved to be published by PeerJ.

Reviewer 1 ·

Basic reporting

I read with interest the paper entitled “Daylight savings time transition and the incidence of femur fractures in the older population”. I agree with the author when they write that this is the first study to investigate the influence of daytime saving time transition on femur fracture incidence. However, authors reported very little information, limiting discussion on data incidence. In my opinion, results should report a complete descriptive analysis of the population. Moreover, I wonder if data on outcomes could be useful.
I would improve the introduction section reporting the relationship between daylight saving time (DST) and acute events such as cardiovascular ones. Moreover, I would link data about trauma and femur fractures at the time of hospitalization. It is not clear if authors included only subjects with femur fracture due to trauma or if they also included subjects who had operation due to different causes.

Experimental design

This is an observational and retrospective study based on International Classification of Diseases (ICD) codes therefore it should be characterized by low sensitivity and specificity, depending on physicians’ ability in codifying hospital procedures and diagnosis could influence the quality of data since different codes are promoted from administrative purpose, addressed to economic rather than research reasons. These studies do not consider clinical or performance parameters and cannot evaluate the events in patients eventually happening after discharge. ICD codes cannot provide information on disease severity, functional status, or reason for intensity of treatment given. In some countries the number of codes that physicians could record is limited. In this paper description about discharge hospital sheets (DHS) compilation is missing. All these points could represent a bias, therefore they should be discussed.
I think that statistical analysis might not be familiar to all readers. Authors should clearly state the statistical analysis steps and the reason for that choice, and they should start from the descriptive analysis. Analysis of events during DST is still a matter of debate and such a problem should be mentioned in the discussion section. I could understand that they evaluated only temporal data, however figure 2 suggests a seasonal pattern of femur fracture that could be eventually confirmed by calculating the observed and expected deaths ratio. Seasonal pattern was never quoted by authors [Ogawa T, Yoshii T, Higuchi M, Morishita S, Fushimi K, Fujiwara T, Okawa A. Seasonality of mortality and in-hospital complications in hip fracture surgery: Retrospective cohort research using a nationwide inpatient database. Geriatr Gerontol Int 2021;21(5):398-403. doi: 10.1111/ggi.14153]. In the results section statistical significance of the multivariate negative binomial regression model is lacking.

Validity of the findings

Finally, I think that discussion section is poor and it does not consider different studies on this item.

Additional comments

Minor remarks
The abbreviation NHDR should be added after the words national Hospital Discharge Register (line 73).
I wonder if the sentence “The most common fall related fractures were included to include to most comprehensive data possible” could be changed to “The most common fall related fractures were considered to include to most comprehensive data possible” (line 152-153).

Reviewer 2 ·

Basic reporting

no comment

Experimental design

no comment

Validity of the findings

no comment

Additional comments

Dear Editor,
I read the paper entitled “Daylight savings time transition and the incidence of femur fractures in the older population”. This is the first study to investigate the influence of daytime saving time transition (DST) on post-traumatic or spontaneous fracture incidence. This is an observational and retrospective study based on International Classification of Diseases (ICD-10) codes on a total of 110 725 femur fractures in Finland.
I have minor criticism on this article in different article sections:
- Introduction: I suggest to improve the introduction section reporting the relationship between DST and cardiovascular events that represent the principal medical conditions related to DST (Manfredini R, et al. J Clin Med. 2019 Mar 23;8(3) - Manfredini R et al. Intern Emerg Med. 2019; 14: 1185–7).
- Materials & Methods: authors evaluated about 110000 femur fractures, but it is not clear if authors included only subjects with femur fracture due to trauma and excluded fracture related to neoplastic condition or if they also included subjects who had operation due to femur fracture. Also authors reported very little information, limiting discussion on data incidence (1997 – 2018). In my opinion, results should report a complete descriptive analysis of the population, in term of comorbidity (such as a comorbidity score) and outcomes. Outcome of these patients could be a factor with which to compare DST-related hip fractures.
The abbreviation NHDR reported in line 83 – 84 should be explain (represent the abbreviation of National Hospital Discharge Register reported in line 73).

- Results: I think that statistical analysis might not be familiar to all readers. Authors should clearly state the statistical analysis steps and the reason for that choice, and they should start from the descriptive analysis. The multivariable negative binomial regression model showed a significant difference only after a DST transition in the spring. In this section, statistical significance of the multivariate negative binomial regression model should be reported. Also, I suggest to showed a figure with different incidence of femur fracture between week following DST transition in the spring and in the fall.
- Discussion: this section is very small, therefore the authors should increase this important section of the study
- Limitations: this is an observational and retrospective study based on International Classification of Diseases (ICD-10) codes therefore it should be characterized by low sensitivity and specificity, depending on physicians’ ability in codifying hospital procedures and diagnosis could influence the quality of data. This condition represents a bias of this study that should be discussed.
For these conditions, I suggest you to accept with minor revision.
Best regards

---

## Round 0.2 · Minor Revisions

Please re-review the comments by reviewers and address any outstanding concerns. Because two reviews made essentially the same comment regarding NB regression, please consider outlining the need for, steps in, and interpretation of NB and Poisson regression. Thank you.

---

## Round 0.3 · accepted · Accept

Thank you very much for persevering and thoroughly addressing the reviewers comments and concerns.

Reviewer 1 ·

Basic reporting

I have no comments, the basic reporting is clear

Experimental design

I have no comments, the experimental design is well reported

Validity of the findings

I have no comments, the findings are valid

Additional comments

I have only minor remarks
Line 122: please delete "include" and add "includes"
Line 203: please delete "distruption" and add "disruption"